# Healthcare professionals' perceptions and recommendations regarding adolescent vaccinations in Georgia and Tennessee during the COVID-19 pandemic: A qualitative research

Olufunto A. Olusanya[1]*, Brianna White[1], Fauzia Malik[2], Kyra A. Hester[3], Robert L. Davis[1], Robert A. Bednarczyk[3], Arash Shaban-Nejad[1]*

1 Department of Pediatrics, Center for Biomedical Informatics, College of Medicine, University of Tennessee Health Science Center, Memphis, Tennessee, United States of America, 2 Department of Health Policy and Management, Yale School of Public Health, Yale University, New Haven, Connecticut, United States of America, 3 Hubert Department of Global Health, Rollins School of Public Health, Emory University, Atlanta, Georgia, United States of America

* oolusan1@uthsc.edu (OAO); ashabann@uthsc.edu (AS-N)

**Data Availability Statement:** All relevant data are within the paper.

## Abstract

### Introduction

Despite its benefits, HPV vaccine uptake has been historically lower than other recommended adolescent vaccines in the United States (US). While hesitancy and misinformation have threatened vaccinations for many years, the adverse impacts from COVID-19 pandemic on preventive services have been far-reaching.

### Objectives

To explore the perceptions and experiences of adolescent healthcare providers regarding routine vaccination services during the COVID-19 pandemic.

### Methodology

Between December 2020 and May 2021, in-depth qualitative interviews were conducted via Zoom video conferencing among a purposively selected, diverse group of adolescent healthcare providers (n = 16) within 5 healthcare practices in the US southeastern states of Georgia and Tennessee. Audio recordings were transcribed verbatim and analyzed using a rapid qualitative analysis framework. Our analysis was guided by the grounded theory and inductive approach.

### Results

Participants reported that patient-provider communications; effective use of presumptive languaging; provider's continuing education/training; periodic reminders/recall messages; provider's personal conviction on vaccine safety/efficacy; early initiation of HPV vaccination

**Funding:** R.B., R.D., and A.S. received the Grant# 1R37CA234119-01A1 from the National Cancer Institute (NCI). No. The funders had no role in study design, data collection and analysis, decision to publish, or preparation of the manuscript.

**Competing interests:** The authors have declared that no competing interests exist.

series at 9 years; community partnerships with community health navigators/vaccine champions/vaccine advocates; use of standardized forms/prewritten scripts/standard operating protocols for patient-provider interactions; and vaccine promotion through social media, brochures/posters/pamphlets as well as outreaches to schools and churches served as facilitators to adolescent HPV vaccine uptake. Preventive adolescent services were adversely impacted by the COVID-19 pandemic at all practices. Participants highlighted an initial decrease in patients due to the pandemic, while some practices avoided the distribution of vaccine informational materials due to sanitary concerns.

## Conclusion

As part of a larger study, we provided contextual information to refine an intervention package currently being developed to improve adolescent preventive care provision in healthcare practices. Our results could inform the implementation of comprehensive intervention strategies that improve HPV vaccination rates. Additionally, lessons learned (e.g. optimizing patient- provider interactions) could be adopted to expand COVID-19 vaccine acceptance on a sizable scale.

## Introduction

Prophylactic vaccinations remain one of the greatest success stories in public health; they are cost-effective interventions that offer protection against infectious diseases and reduce morbidity and mortality [1]. Due to successful vaccination programs, vaccine-preventable diseases have been controlled or eliminated in the United States (US) [2–4]. In late 2019, the highly transmissible Coronavirus Disease 2019 (COVID-19) caused by the severe acute respiratory syndrome coronavirus 2, or SARS-COV-2, was declared a pandemic by the World Health Organization [5]. More recently, the B.1617.2 (delta) variant of the SARS-COV-2 has resulted in severe outcomes among children/adolescents across the US. At the time of the initial manuscript preparation, as of August 14, 2021, the weekly COVID-19–associated hospitalization rate per 100,000 for children/adolescents was approximately five times the rate recorded on June 26, 2021 [6]. Moreover, community transmission of SARS-COV-2 remained high in the US with fear of uncontrolled spread [7].

Nonetheless, challenges related to the COVID-19 pandemic reach far beyond the disease; disrupting and reversing gains made to routine childhood/adolescent vaccinations in the U.S. [8–10] and globally [11,12]. Specifically, vaccinations are being forestalled by the adverse impacts of pandemic disruptions such as lockdowns, loss of income, and/or lack of health insurance, and other quarantine measures. These disruptions have led to limited healthcare access and services, hindered effective patient-provider communications, and increased parental vaccine hesitancy and reluctance [13]. The vaccines that are currently approved for adolescents in the US include tetanus, diphtheria, and acellular pertussis (Tdap); meningococcal conjugate (MenACWY); human papillomavirus (HPV); influenza; and COVID-19 [14]. However, between 2019 and 2021, the global prevalence of children vaccinated with Tdap (key indicators for vaccination coverage) fell by 5%. [12] Moreover, global HPV vaccination coverage reduced by more than 25% over the same period [12]. The pandemic also caused a consequential 42% decline in all Vaccines for Children Program (VFC) vaccine orders from providers [10,15]. Accordingly, Daniels and colleagues projected tens of thousands of additional cases of

HPV-associated genital warts and precancerous cervical lesions following reduced HPV vaccine coverage due to the COVID-19 pandemic [15].

Although the HPV vaccine is highly protective against most HPV-associated cancers–cervical, oropharyngeal, anal, penile, and vaginal [16]–the implementation of strategies to promote HPV vaccination coverage among adolescents has been a daunting task. Before the pandemic, HPV vaccine uptake was historically lower than uptake rates recorded for the two school-mandated vaccines for adolescents—Tdap and MenACWY. In 2019, HPV vaccine completion rates for adolescents (13–17 years) were estimated at 49.7% and 43.0% in Georgia and Tennessee, respectively, versus the national coverage rate recorded at 54.2% [17,18]. Similarly, completed COVID-19 vaccine primary series that were administered in Georgia and Tennessee were lower at 18.8% and 16.8%, respectively, compared to the national coverage at 31.9% as of manuscript preparation on July 31, 2021, for adolescents (12–17 years) [19].

The COVID-19 pandemic has worsened vaccine mis- and disinformation and exacerbated the public's mistrust in vaccines, government officials, and public health institutions. However, the COVID-19 vaccine is recommended as effective and safe for preventing serious COVID-19 infections, hospitalizations, and deaths [20]. Studies have reported that a lack of healthcare providers' recommendations, low level of knowledge, parental concerns about vaccine safety/ efficacy, and religious/philosophical beliefs have hindered adolescent vaccination uptake [21]. Healthcare professionals play a central role in maintaining public trust in vaccination programs and are best positioned to address parental vaccine hesitancy/refusal/delay. Studies also showed that healthcare providers' recommendations strongly predicted immunization uptake [22–25]. It is pertinent that vaccination viewpoints, perspectives, and practices are examined among healthcare professionals within the context of the pandemic disruptions to healthcare systems. Herein, we explored the perceptions and experiences of adolescent healthcare providers in Georgia and Tennessee as they delivered preventive health services during the COVID-19 pandemic. As part of a larger vaccine promotion study, our qualitative findings will inform the implementation of comprehensive strategies to address suboptimal adolescent vaccination (e.g. HPV, COVID-19, etc.) delivery systems in pediatric healthcare practices across the states of Georgia and Tennessee.

## Materials and methods

In-depth semi-structured qualitative interviews were conducted among adolescent healthcare professionals in Georgia and Tennessee between December 2020 and May 2021. The Institutional Review Boards (IRB) at Emory University and The University of Tennessee Health Science Center (UTHSC) approved this study.

## Study area and population

Georgia and Tennessee, located in the southeastern region of the US, are comparable in demographic representations and geographic variability for HPV vaccine completion rates. In both states, adolescents residing in Metropolitan Statistical Areas (MSA), non-central, and non-MSA were less likely to have completed their HPV vaccine doses. Our study targeted community-level pediatric healthcare practices where adolescent preventive services were available and offered to 11- and 12-year-olds. Overall, five healthcare practices participated in our study; four from Tennessee and one from Georgia. Our inclusion criteria consisted of physicians, nurses, nurse practitioners, front desk staff, medical assistants, and practice managers/ administrators.

## Recruitment and data collection

We utilized a purposive sampling strategy where healthcare practices from two states, Tennessee (n = 14) and Georgia (n = 2) field sites, respectively, were identified and selected through referrals from professional organizations, community partners, healthcare providers, and internet searches. Healthcare practice managers and providers were contacted using recruitment emails and a cold-call system. Recruitment emails contained the following: study overview, written consent form, Qualtrics link to obtain demographic information, and request to schedule a qualitative interview. The consent form contained information on participants' right to refuse/withdraw from participation, as well as data privacy and protection protocols. Each participant received a $50 gift card after the completion of their qualitative interview.

## In-depth qualitative interviews with healthcare professionals

The interview guide was developed by a multi-disciplinary team of researchers from Emory and Yale Universities with expertise in implementing evidence-based interventions that targeted practice-, provider- and patient-level barriers to vaccine uptake. The same qualitative guide was used for each interview to ensure similar topics were discussed with each participant. Questions were phrased in a clear and neutral tone, avoiding leading language and unnecessary jargon. Moreover, questions were developed to limit transactional question-and-answer interactions and instead, stimulate open-ended, unfolding responses. To supplement primary questions, predetermined and practiced targeted follow-up probes were used to provide clarification as needed.

The qualitative interview guide comprised six sections with questions exploring the following pre-determined themes: (1) participant background and perspectives on adolescent preventive care, (2) description of both routine and preventive care visits within the healthcare practice, (3) staff communications and activities including training, standards, quality improvement, and monitoring activities, and areas of improvement/success, (4) procedures regarding the provision of adolescent vaccinations, (5) communication and recommendations with/to adolescents and their parents, (6) impact of COVID-19 on practice, and (7) recommendations for programs to improve HPV vaccine uptake.

Face-to-face interviews are considered the gold standard for data collection in qualitative research, however, the COVID-19 pandemic has rapidly expedited the use of alternate data collection techniques i.e., online, video, and telephone. A team well-trained in qualitative methods conducted the interviews via the video conferencing platform, Zoom. Interviews were scheduled to suit participants' availability and conducted in a private, quiet area with reliable technology and stable internet. In addition to obtaining written informed consent, verbal consent was also taken before audio/video recordings started in Zoom.

## Data management and rapid qualitative analysis

All interviews were audio recorded. Data were uploaded and stored in password-protected, HIPAA-compliant university data storage cloud platforms. Audio recordings were transcribed verbatim. All transcripts were de-identified to ensure participants' privacy/confidentiality and were accessible to only the research team.

A team-based rapid thematic analysis framework was used to collect data and conduct a narrative thematic analysis of transcripts. Quantitative demographic data was utilized during the analysis process; however, solely for comparative use. A detailed codebook and code definitions were developed to elicit significant details from the transcripts such as "Background & Context", "Adolescent Care Visits", "Staff Communications and Activities", "Adolescent Vaccination", and "Provider/Clinical Staff Addressing Hesitance", to name a few. The thematic

categories and coding scheme were further refined through a series of iterative cycles to define key subthemes and themes as used in qualitative analysis [26]. To establish rigor of the analysis, a diverse range of viewpoints and experiences from diverse categories of participants (i.e., physicians, nurses, medical assistants (MAs), practice managers, administrative staff, and front desk personnel) were compared and included [27]. We verified code saturation and informational redundancy had occurred when (1) no new codes were identified, and (2) researchers repeatedly heard the same themes from prior qualitative interviews. The dependability of our analysis was established using one consistent interview guide and thorough documentation of all decisions reached during data collection and analysis [28]. Study credibility and confirmability were enforced by multiple independent coders, investigator triangulation, use of participants' verbatim quotes for analysis, and comparison of participants' accounts to ensure that a wide range of perspectives was represented [28]. Final thematic groupings were further assessed by two members of the research team to compare for consistency. Overall, our theme/subtheme framework consisted of seven domains as shown in Appendix A. Our analysis was guided by the grounded theory and inductive approach.

## Results

Overall, 16 participants completed in-depth qualitative interviews. Recruited participants included a role-diverse pool of preventive healthcare providers across both states who were pediatricians, internal medicine physicians, nurse practitioners, MAs, practice managers, administrative staff, and front desk personnel. Most participants were female (75.0%), within 26–35 years of age (45.5%), spent 6–10 years in current practice (36.6%), and had children (90.1%). Participants identified as medical doctors (56.3%), registered nurses (12.5%), MAs (18.8%), and front desk/administrative staff (12.5%). The median interview completion time was approximately 46 minutes.

### Typical preventive adolescent care visits within healthcare practices

On arrival, adolescents and their parents/caregivers checked in at the front desk to complete paperwork and health insurance verification. Thereafter, patients were transferred to the examination room to have their vitals and mental health assessed. Next, a provider performed a physical examination, obtained medical history, and ordered due vaccinations. Most practices had standing orders for nurses-only visits which permitted nurses or MAs to administer vaccines in accordance with CDC and American Association of Pediatrics (AAP) immunization guidelines.

During wellness visits, adolescents' vision, hearing, weight, height, temperature, blood pressure, and heart rate were assessed by the triage nurses or MAs. Validated pediatric symptom questionnaires and checklists were administered in the waiting area or examination room by the nurses/MAs to assess substance use, behavior health, allergies, diet change, etc. Moreover, screenings for sexually transmitted infections (STIs), cholesterol, scoliosis, etc. were ordered at most healthcare practices. Sometimes, parents/caregivers were asked to leave the examination room while physicians conducted their patient clinical assessment. Patients' vaccination statuses were checked using CDC and AAP guidelines.

Typical wait times for adolescents/caregivers were reported to differ across participating healthcare practices. Typical wait times reported by participants ranged between 5 to 90 minutes and were contingent upon how busy the clinics were. Some participants perceived lengthy waiting times to be a source of concern:

"*My location biggest flaw is the wait period for the patients. If everybody shows up at one time, then we can fill up the waiting room and see everybody else in the car. And they are in*

*the car until we're able to get available rooms, which can be anywhere from instant to 45 minutes."*

Based on the protocol in some practices, adolescents/caregivers had access to educational materials on vaccinations (i.e., brochures, posters, pamphlets) while waiting in the lounge area or examination room.

*"Of course course, we have the pamphlet with the pictures on it—so my high schoolers. . ..You know, you tell them about it, and then once you show the boys the picture. They want the vaccine, especially at warts, they're like. . . oh, uh. Yeah. . .please give me that one."*

**Patient-provider communications and adolescent vaccinations.** Physicians were mostly responsible for disseminating vaccine information by recommending all 3 vaccines (i.e., Tdap, MenACWY, HPV) during the same visit. Parents were sometimes given educational materials during initial vaccine discussions to review more thoroughly at home. Generally, patient-provider communications include information on HPV vaccine efficacy against HPV-associated cancers and genital warts. Several participants noted that Tdap and MenACWY were recommended as required by state health and school officials, while HPV vaccine was described as optional yet highly recommended for the prevention of HPV-associated infections/cancers. Occasionally, parents/caregivers were interested in the healthcare provider's personal opinions regarding vaccinations. Overall, many participants highlighted the importance of building trusting relationships and rapport with parents/patients using personal experiences and addressing parental fears/concerns.

*"They always ask me personal experience. Do you vaccinate your kids? Would you let your kids get this? And I'm like hands down, I trust my doctor . . .So, if my doctor says this is a good vaccination, this is good for my kids, I'm going for it"*

and *"Telling people what I did personally, what I did with my kids when it comes to vaccinations, specifically and allowing them to answer their questions and don't strong arm them."*

**Impact of COVID-19 on pediatric practices.** All practices interviewed implemented preventive measures due to COVID-19. Practices either required patients/caregivers to check in online rather than visit the front desk or had staff perform temperature checks and screening questionnaires. Healthcare practices typically distribute information from CDC promoting vaccination, however, some practices avoided the distribution of physical materials due to sanitary concerns related to the pandemic. While participants highlighted an initial decrease in patients due to the pandemic, several practices reported facilitating catch-up vaccinations for those with missed opportunities. Interestingly, a few participants expressed concerns that controversies surrounding the COVID-19 vaccine safety could impact the receipt of other vaccinations.

**Health record review, appointment reminder systems, and community health navigators.** During provider encounters, a one-page checklist was used as a guide along with online records from the electronic medical records (EMR) to determine what preventive care services were due. Following visits at some practices, parents were sent "ticklers" or generic reminder messages to schedule subsequent appointments. These reminders/recall systems offered information on the patient's appointment date/time and doctor's name/location. Sometimes, these

reminders delivered prompts in the form of phone calls, text messages, and emails reminding parents to schedule appointments by phone calls or through the online EMR portal.

After the first HPV vaccine, most reminder/recall systems utilized patients' subsequent annual visit or an interim visit for a chronic condition (standardized 6-month reminders did not exist) for the second HPV dose. A few healthcare practices reportedly sent reminders to 9-year-old patients (as opposed to 11- and 12-years) which resulted in increased HPV vaccine uptake. One participant mentioned their use of "community health navigators" who gauged vaccination success/failure rates through patients' follow-up visits and vaccination rates.

## Staff communications and activities around preventive healthcare

**Staff training and use of standard operating protocols.**   Front desk/administrative staff did not typically provide recommendations to parents; most non-clinical staff had not undergone formal standardized training. One participant, however, described vaccine communication strategies training for front desk personnel to improve HPV vaccination uptake at their healthcare practice. Another participant cited a 6-month training received via the state chapter of the AAP program. A handful of practices reported annual VFC or Objective Structured Clinical Examination (OSCE) training for nurses while resident physicians received Continuity Clinic Curriculum and Continuing Medical Education (CME) courses on vaccines. Additionally, most practices did not utilize standardized forms/sheets, prewritten scripts, or standard operating protocols (SOP) to ensure consistent and standard messaging to parents and patients.

"*I don't know that there's any. . . standardized, um, communication that our nurses and MAs do. I think it probably varies a lot from person to person. Sometimes I will hear them say, you're due for vaccines today but other times I won't, um, so, I'm not sure that there's any standardized language or communication that's really in place right now.*"

As one participant noted, instead, they were given a set of "dos and don'ts" for their position/role. One participant referenced a manager who frequently presented "pop quizzes" and served refreshers to ensure the front desk staff was up to date on relevant information. Overall, most in-depth communications related to vaccinations were referred to and conducted by nurses and physicians.

**Quality improvement and monitoring activities.**   Quality improvement measures were implemented by clinical/nonclinical staff to improve vaccination rates. A few participants indicated being members of the Family Interaction Training (FIT) program through Tennessee's AAP and The Children's Care Network (TCCN) where vaccination rates and frequency of wellness visits were assessed. Another participant utilized PDSA (Plan, Do, Study, Act) goals. One practice mentioned working with the Patient-Centered Medical Home (PCMH), Tenn-Care, and HEDIS to meet the criteria for vaccination services. Several participants expressed that it was unrealistic to anticipate 100% vaccination rates.

Overall, most did not work with AFIX and HEDIS. Those who did, expressed difficulty in sustaining the program. Some participants were not fully aware of any quality improvement measures for vaccine compliance within their healthcare practices.

## Parent/Adolescent vaccination concerns and addressing hesitancy

**Parent/Caregiver and adolescent vaccine concerns.**   Participants expressed that some parent concerns were regarding HPV vaccine safety and content with vaccines "*messing with their (children's) DNA*", and vaccines being "*made using pieces of a baby.*" Parents also

frequently voiced fears/concerns about short- and long-term side effects, pain at injection sites, and religious beliefs about vaccines; additional inquiries were made by parents regarding autoimmune diseases, infertility, menstrual disorders, and death. Notably, parents' concerns were rarely related to MenACWY or Tdap vaccines; however, these were primarily aimed at the HPV vaccines.

> "*With Tdap and Meningitis, they'll ask, will their arm be sore? Will they have a fever? Whereas if it's Gardasil (HPV vaccine), they're asking autoimmune stuff, fertility problems, problems with your period. . ..*"

> and "*Most people accept the fact that they're getting Menactra . . . but for the Gardasil and the flu. Most of them are, 'well, Gramma said not to. So, I'm not going to. . .*"

Parents also conveyed concerns regarding "unverified" associations between HPV vaccine and sexual promiscuity/premarital sex in their children. Many parents had misconceptions that their children didn't require HPV vaccinations because they were not sexually active:

> "*Some people think that because it's a sexually transmitted infection, that their kids aren't having sex and [getting vaccinated for HPV] might give them permission to have sex, and we're not even going to need that because they're gonna wait until they're married.*"

One participant reported addressing this misconception with the following:

> "*Unfortunately, not all people choose their sexual activity. . .. And so, I say—even if—it's usually a girl—she or he does not have sex until they're married, and you have no idea what their partner's history will be. Or, unfortunately, not all marriages are monogamous. . ..whereas you could protect yourself because you don't always have control over your exposure.*"

Except for the pain at injection sites following vaccinations, most adolescent patients did not express concerns. As explained by participants, most adolescents perceived their parent's consent/authorization was required to get vaccinated; therefore, adolescents usually complied with whatever decisions their parent(s)/caregiver(s) made.

**Addressing parental vaccine hesitancy.** A number of participants cited increased HPV vaccine compliance when providers announced or used the phrase, "*You are due for Tdap, Meningococcal, and HPV vaccines today.*" In other words, vaccines were not recommended solely based on school-entry requirements. This type of messaging, known as presumptive languaging, was reported to be effective as parents were often agreeable to the receipt of all 3 vaccines. Other participants reported addressing parental hesitancy through in-depth discussions on vaccine safety, effectiveness, importance, content, and possible side effects. Such interactions were perceived to encourage discussions that clarified parents' questions/concerns as well as facilitate decision-making and behavior change. Some participants expressed their personal convictions and experience on vaccine safety/benefits to address parental vaccine hesitancy: "*. . ..I vaccinated them on time as soon as I could because I know that the benefits outweigh the risks that I plan on giving my son this vaccine. . . .*"

In addition, most healthcare practices had policies requiring patients to be fully vaccinated. Families were given the time/space to make informed decisions regarding vaccinations; however, those who continued to refuse vaccinations were asked to complete a vaccine refusal form and were referred to another clinic.

*"Our expectation is that if you are a patient in our practice, that if your child is not up-to-date on immunizations by the time you've been with us for 2 years, that we dismiss you from our practice. . . .We don't officially offer any alternative vaccine schedules."*

Some healthcare providers worked with parents to space out vaccinations and ensure children received all their vaccines. Others commenced HPV vaccine discussions at the earlier age of 9 years.

*"You know the recommendation is 12 or 13 years, maybe 11. But because it's indicated down to nine we have recently started [vaccinating for HPV] at our nine-year-old checkups, for one, because our. . . population is kind of a high-risk population."*

**Opportunities for information sharing.** All participants indicated their data, vaccination statistics, and general metrics were captured and generated through the EMR systems allowing patients/parents and clinical/non-clinical staff to immediately access, view, track records/vaccination status, and provide feedback. Most practices had official websites and social media accounts for disseminating information. Others had communication exchange mechanisms whereby specialized clinics within a healthcare practice made referrals. Others proposed the use of care coordinators, social workers, and case managers to share information and follow up with caregivers. One participant recommended the use of school outreaches/campaigns to adolescents.

*"Educating the middle schoolers and. . .. then sending that information home to the parents cause obviously they're the ones that ultimately making the decision."*

## Discussion

HPV vaccination programs targeted at eligible individuals are highly effective and shown to reduce the risk of HPV-associated infections and cancers. Effective personalized patient-provider communications are crucial for facilitating vaccine acceptance and addressing parental hesitancy. Healthcare providers are uniquely positioned to provide recommendations based on CDC and AAP vaccine guidelines to increase vaccination rates. Using in-depth qualitative interviews, this study explored the perspectives, opinions, and experiences of healthcare professionals who interacted closely with adolescents to provide routine vaccination services during the COVID-19 pandemic. This study could facilitate vaccine implementation and interventions that mitigate the adverse impacts of the pandemic on routine adolescent immunizations by proposing the following: (1) prioritize HPV vaccine equity, access, and affordability; (2) overcome barriers to patient-parent-provider interactions; (3) optimize patient visits to support increased vaccination and address missed opportunities; (4) support education and training to increase providers' confidence; (5) utilize big data analytics to monitor public trust/sentiments and identify trends in vaccine uptake; and (6) implement policies that facilitate increased vaccine uptake.

Standing orders, telemedicine video conferencing, and administration of vaccines through outdoor/curbside/drive-through services are crucial to prioritize equitable access to HPV vaccines, capture missed opportunities and ensure catch-up vaccinations [13,29]. Moreover, participants made recommendations to incorporate pertinent vaccine information (i.e., waived vaccine costs for under-/uninsured) in reminder/recall messages, thus facilitating vaccination decision-making before appointment visits. Previous research shows that adolescents have

expressed the desire to acquire HPV vaccine knowledge and be involved in the vaccination decision-making process [21,30]. To promote information-seeking and decision-making in today's digital age, educational marketing/outreach through social media platforms, televised advertisements, radio commercials, and billboards can be utilized to disseminate vaccine knowledge and awareness among adolescents and their parents/caregivers. Vaccine informational materials can be offered to parents/adolescents via partnerships with healthcare practices; religious institutions and schools (i.e., during health education and sciences classes). Other best practices proposed by participants included the configuration of EMRs to send HPV vaccination prompts/alerts to providers when patients clock 9 years old as well as the use of community health navigators, vaccine champions, and vaccine advocates to promote vaccine benefits, particularly to high-risk populations.

To address barriers affecting patient-parent-provider interactions, several participants proposed the effectiveness of presumptive languaging [31] for increasing vaccine compliance within their healthcare practices. These participants indicated that when used by providers, the presumptive approach was linked to improved vaccine uptake when compared to conversational languaging [31]. Moreover, the application of motivational interview techniques (i.e., acceptance, compassion, collaboration, etc.) during patient-provider interactions are shown to be effective at enabling trusting relationships and addressing hesitancy [32,33]. Importantly, patient-provider interactions should encourage patient engagement and support shared decision-making between adolescents and their parents/caregivers. Moreover, when commenced at an earlier patient's age (i.e., 9 years old) parent-provider interactions are less likely to focus on the patient's sexual activity as well as offer adequate time to address parents' concerns. Our study indicated that personal connections and trust between providers and parents/caregivers facilitated effective vaccination discussions. Accordingly, healthcare providers are to use their personal experiences when discussing vaccines and offering strong recommendations to families.

Some participants reported lengthy wait times for patients; as such, patients' visits should be optimized using informational sources such as brochures, posters, or pamphlets to increase vaccine uptake and enable healthy behavior (i.e., safe sex practices, smoking cessation, etc.) Also, providers should offer timely consultations to reduce wait times and improve patient satisfaction. Overall, collaborative, and effective vaccine uptake strategies that optimize patients' visits should be promoted among both clinical and non-clinical healthcare providers. Moreover, addressing barriers to vaccination quality improvement programs within healthcare settings should be prioritized through implementing simple, efficient, standard, and automatic systems that lessen providers' burden [34]. Furthermore, to optimize patients' clinical encounters, training/educational courses should aim to increase health providers' confidence to address parents' fears/concerns and underscore vaccine benefit/safety/efficacy. While most vaccine recommendations were offered by physicians/nurses, another focal point of contact for information sharing (i.e., front desk staff and MAs) should be effectively utilized. Continuing education programs and periodic training should be delivered to expand knowledge and expertise among clinical/non-clinal staff.

Certain groups and populations have expressed their distrust in HPV vaccines because these are erroneously perceived to be unsafe, ineffective, and increase the propensity for sexual exposure among adolescents. Besides, parental hesitancy has been linked to other vaccine mis- and disinformation (i.e., autoimmune diseases, post-vaccination mortalities, etc.) which have been overwhelmingly debunked by scientific evidence [35–38]. Therefore, it is imperative that artificial intelligence is utilized to examine prevalence and trends in HPV vaccine uptake, assess social determinants of health (SDoH) and inequalities/disparities [39] as well as monitor public trust/sentiments [40,41] in vaccines and public institutions. Moreover, provider

speaking prompts/prewritten scripts/standard operating protocols [36] should be adopted to dispel misinformation and ensure that consistent, unambiguous information is delivered to parents/caregivers. Lastly, policies that facilitate increased HPV vaccine rollout and uptake ought to be implemented such as minor consent laws which permit vaccinations against HPV and COVID-19 among adolescents (12–17 years) in the absence of parental consent [42]. Likewise, school-entry policies can include requirements for the HPV and COVID-19 vaccines to contribute to high vaccination rates among students [43].

Some study limitations should be considered in unison with our findings. This study was conducted among 16 healthcare professionals that were affiliated with 5 healthcare practices within 2 US southeastern states. Therefore, our study results are not generalizable to the states and the US and might not fully encompass underlying complexities impacting vaccination services at other locations. Also, due to constraints related to the ongoing COVID-19 pandemic, our recruitment process and qualitative interviews were conducted via Zoom video call as opposed to in person. Nevertheless, quantifiable differences between video calls and in-person interviews have been described as only marginal [44,45]. Despite these limitations, qualitative data were obtained from a diverse group of healthcare professionals. Overall, our study results could inform the application of behavior change interventions among providers, patients, and parents/caregivers to facilitate optimal adolescent preventive care services and increase vaccination rates. Also, importantly, lessons learned, and vaccine promotion strategies used to facilitate adolescent HPV vaccine uptake could be adopted to improve COVID-19 vaccinations, and vice versa.

## Supporting information

**S1 Fig. Themes and subthemes used for rapid qualitative analysis.**
(TIF)

## Acknowledgments

We thank Hailey Hwang, Morgan Fleming, and Adrian King for their roles in the data collection process and program administration. We also extend our profound gratitude to all healthcare professionals who participated in our study.

## Author Contributions

**Conceptualization:** Fauzia Malik, Robert L. Davis, Robert A. Bednarczyk, Arash Shaban-Nejad.

**Data curation:** Olufunto A. Olusanya, Brianna White.

**Formal analysis:** Olufunto A. Olusanya, Brianna White.

**Funding acquisition:** Fauzia Malik, Robert L. Davis, Robert A. Bednarczyk, Arash Shaban-Nejad.

**Methodology:** Olufunto A. Olusanya, Brianna White, Fauzia Malik, Robert L. Davis, Robert A. Bednarczyk, Arash Shaban-Nejad.

**Project administration:** Olufunto A. Olusanya, Brianna White.

**Supervision:** Olufunto A. Olusanya, Fauzia Malik, Robert L. Davis, Robert A. Bednarczyk, Arash Shaban-Nejad.

**Validation:** Olufunto A. Olusanya.

**Writing – original draft:** Olufunto A. Olusanya, Brianna White.

**Writing – review & editing:** Olufunto A. Olusanya, Brianna White, Fauzia Malik, Kyra A. Hester, Robert L. Davis, Robert A. Bednarczyk, Arash Shaban-Nejad.

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
