## [Decision Letter · Decision Letter 0]

12 Sep 2022

PONE-D-21-35597Healthcare Professionals’ Perceptions and Recommendations regarding Adolescent Vaccinations in Georgia and Tennessee during the COVID-19 Pandemic: A Qualitative ResearchPLOS ONE

Dear Dr. Olusanya,

Thank you for submitting your manuscript to PLOS ONE. After careful consideration, we feel that it has merit but does not fully meet PLOS ONE’s publication criteria as it currently stands. Therefore, we invite you to submit a revised version of the manuscript that addresses the points raised during the review process.

We look forward to receiving your revised manuscript.

Kind regards,

Florian Fischer

Academic Editor

PLOS ONE

Journal Requirements:

Reviewers' comments:

Reviewer's Responses to Questions

**Comments to the Author**

1. Is the manuscript technically sound, and do the data support the conclusions?

Reviewer #1: Yes

Reviewer #2: Partly

2. Has the statistical analysis been performed appropriately and rigorously? 

Reviewer #1: Yes

Reviewer #2: N/A

3. Have the authors made all data underlying the findings in their manuscript fully available?

Reviewer #1: Yes

Reviewer #2: Yes

4. Is the manuscript presented in an intelligible fashion and written in standard English?

Reviewer #1: Yes

Reviewer #2: No

5. Review Comments to the Author

Reviewer #1: 1. In the Introduction section, author(s) write “Nonetheless, challenges related to the COVID-19 pandemic reach far beyond the disease; disrupting and reversing gains made to routine childhood/adolescent vaccinations.”, however, there seems to be a lack of evidence in the following to support this view. It would be good to provide more data or findings from published studies to support this argument.

2. In the Introduction section, more information about the previous study findings should be presented. A thorough literature review is needed to provide a rationale for conducting this study.

3. Face-to-face interviews and focus groups have been the norm for data collection in the qualitative research. During the period of COVID-19 pandemic, online methods appear to increase the likelihood of obtaining the desired sample, but responses are usually shorter, less contextual information is obtained, and lower relationship satisfaction may occur. How did the authors ensure the quality of the interviews online?

4. How did the authors identify the interview topics and was there a corresponding theoretical framework? How was the interview guide determined? Explanation is needed.

5. Did the interview data included in this manuscript indicate that there is already saturation? The author needs to explain this.

6.Having some bullet points on how to address the new perceptions or recommendations of the medical staff during the COVID-19 epidemic would be beneficial for the Discussion and the entire manuscript.

7. The authors need to explain what new information they have explored in this study? What future implications does this study have for the improvement or enhancement of vaccination services for adolescents locally and globally?

Reviewer #2: This is a potentially interesting paper that involves in-depth interviews of healthcare professionals’ perceptions and recommendations regarding adolescent vaccines. This is an important topic with limited studies in the literature. However, the paper requires a substantial revision with assistance from one of the senior authors. Some issues are small, such as “Numerous devastating contagious diseases such as Smallpox have been completely eradicated in the United States”. Eradication is global. Many diseases have been controlled or eliminated in the US. More broadly, the paper needs to be substantially reduced in length to be more succinct. There also needs to be much more detail in the data analysis section including the qualitative analysis framework that was used, and how it was used. This section requires review by a qualitative methodologist. The Bright Future Immunization Schedule requires a description as it is unlikely to be familiar to most readers.

6. PLOS authors have the option to publish the peer review history of their article (what does this mean?). If published, this will include your full peer review and any attached files.

Reviewer #1: No

Reviewer #2: No

---

## [Author Response · Author response to Decision Letter 0]

27 Oct 2022

PLOS ONE Manuscript: Response to Reviewer Comments

Editorial comments:

and

Response: We thank the editor and reviewers for their feedback. We appreciate the opportunity to reflect on and improve the manuscript. We have modified our manuscript to reflect PLOS ONE’s style requirements. 

Comments to the Author

1. Is the manuscript technically sound, and do the data support the conclusions?

Reviewer #1: Yes

Reviewer #2: Partly

2. Has the statistical analysis been performed appropriately and rigorously?

Reviewer #1: Yes

Reviewer #2: N/A

3. Have the authors made all data underlying the findings in their manuscript fully available?

The PLOS Data policy requires authors to make all data underlying the findings described in their manuscript fully available without restriction, with rare exception (please refer to the Data Availability Statement in the manuscript PDF file). The data should be provided as part of the manuscript or its supporting information, or deposited to a public repository. For example, in addition to summary statistics, the data points behind means, medians, and variance measures should be available. If there are restrictions on publicly sharing data—e.g. participant privacy or use of data from a third party—those must be specified.

Reviewer #1: Yes

Reviewer #2: Yes

4. Is the manuscript presented in an intelligible fashion and written in standard English?

Reviewer #1: Yes

Reviewer #2: No

Detailed Comments:

Reviewer #1: 

1. In the Introduction section, author(s) write “Nonetheless, challenges related to the COVID-19 pandemic reach far beyond the disease; disrupting and reversing gains made to routine childhood/adolescent vaccinations.”, however, there seems to be a lack of evidence in the following to support this view. It would be good to provide more data or findings from published studies to support this argument.

Response: We appreciate this feedback. We have now cited scientific evidence which show a decline in childhood vaccinations due to the COVID-19 pandemic both in the US and globally.

2. In the Introduction section, more information about the previous study findings should be presented. A thorough literature review is needed to provide a rationale for conducting this study.

Response: We appreciate the reviewer’s attention to this issue. The last 3 paragraphs in the introduction clearly explain our rationale for conducting this study. Moreover, more information and literature review from previous study findings are presented and compared to our results in the discussion section.

3. Face-to-face interviews and focus groups have been the norm for data collection in the qualitative research. During the period of COVID-19 pandemic, online methods appear to increase the likelihood of obtaining the desired sample, but responses are usually shorter, less contextual information is obtained, and lower relationship satisfaction may occur. How did the authors ensure the quality of the interviews online?

Response: We understand and appreciate this concern. In this paper, we have cited studies that show that quantifiable differences between video calls and in-person interviews are marginal. Also, as specified in our paper, we took definite steps to address the limitations of online data collection. 

4. How did the authors identify the interview topics and was there a corresponding theoretical framework? How was the interview guide determined? Explanation is needed.

Response: We appreciate the reviewer’s feedback. In response, we have added an explanation to our Materials and Methods section. Our response can be found in the “In-depth qualitative interviews with healthcare professionals” section above.

5. Did the interview data included in this manuscript indicate that there is already saturation? The author needs to explain this.

Response: We have indicated in our manuscript that data saturation occurred. We have also explained how we (researchers) determined that data satration had occurred. 

6. Having some bullet points on how to address the new perceptions or recommendations of the medical staff during the COVID-19 epidemic would be beneficial for the Discussion and the entire manuscript.

Response: We thank the reviewer for their feedback and agree that the discussions/recommendations section should be organized using bullet points. As a result, we have presented our recommendations using a more logical and organized approach. 

7. The authors need to explain what new information they have explored in this study? What future implications does this study have for the improvement or enhancement of vaccination services for adolescents locally and globally?

Response: As indicated in the last paragraph of the introduction section, we have explained the new information explored in this study, “It is pertinent that vaccination viewpoints, perspectives, and practices are examined among healthcare professionals within the context of the pandemic disruptions to healthcare systems.” Also, in the last paragraph of the discussion, we have also highlighted the future implications this study has on improving vaccination services. 

Reviewer #2: 

1. This is a potentially interesting paper that involves in-depth interviews of healthcare professionals’ perceptions and recommendations regarding adolescent vaccines. This is an important topic with limited studies in the literature. However, the paper requires a substantial revision with assistance from one of the senior authors. Some issues are small, such as:

a. “Numerous devastating contagious diseases such as Smallpox have been completely eradicated in the United States”. Eradication is global. Many diseases have been controlled or eliminated in the US. 

Response: We thank the reviewer We have edited/revised the sentence based on the reviewer’s suggestions. 

b. More broadly, the paper needs to be substantially reduced in length to be more succinct. 

Response: We appreciate the reviewer’s comment. We have reduced the paper’s length. However, we have also incorporated missing/additional information that was requested by the second reviewer. 

c. There also needs to be much more detail in the data analysis section including the qualitative analysis framework that was used, and how it was used. This section requires review by a qualitative methodologist. 

Response: We appreciate the reviewer’s feedback. In response, we have added additional detail to highlight the qualitative analysis framework. Our robust responses can be found throughout the “Materials and Methods” and “Data Analysis” sections.

d. The Bright Future Immunization Schedule requires a description as it is unlikely to be familiar to most readers.

Response: We have restricted the use of our immunization schedules to CDC and AAP guidelines. As a result, we have deleted the Bright Future Schedule in the manuscript.

---

## [Decision Letter · Decision Letter 1]

3 Nov 2022

Healthcare Professionals’ Perceptions and Recommendations regarding Adolescent Vaccinations in Georgia and Tennessee during the COVID-19 Pandemic: A Qualitative Research

PONE-D-21-35597R1

Dear Dr. Olusanya,

We’re pleased to inform you that your manuscript has been judged scientifically suitable for publication and will be formally accepted for publication once it meets all outstanding technical requirements.

Kind regards,

Florian Fischer

Academic Editor

PLOS ONE

Additional Editor Comments (optional):

Reviewers' comments:

Reviewer's Responses to Questions

**Comments to the Author**

1. If the authors have adequately addressed your comments raised in a previous round of review and you feel that this manuscript is now acceptable for publication, you may indicate that here to bypass the “Comments to the Author” section, enter your conflict of interest statement in the “Confidential to Editor” section, and submit your "Accept" recommendation.

Reviewer #1: All comments have been addressed

Reviewer #2: All comments have been addressed

2. Is the manuscript technically sound, and do the data support the conclusions?

Reviewer #1: Yes

Reviewer #2: Yes

3. Has the statistical analysis been performed appropriately and rigorously? 

Reviewer #1: Yes

Reviewer #2: N/A

4. Have the authors made all data underlying the findings in their manuscript fully available?

Reviewer #1: Yes

Reviewer #2: Yes

5. Is the manuscript presented in an intelligible fashion and written in standard English?

Reviewer #1: Yes

Reviewer #2: Yes

6. Review Comments to the Author

Reviewer #1: The manuscript has been revised as suggested. I recommend publication.

Reviewer #2: The revision adequately addresses comments. The authors clearly made reasonable efforts to consider reviewer comments and revise accordingly.

7. PLOS authors have the option to publish the peer review history of their article (what does this mean?). If published, this will include your full peer review and any attached files.

Reviewer #1: No

Reviewer #2: No

---

## [Editor Report · Acceptance letter]

10 Nov 2022

PONE-D-21-35597R1 

Healthcare professionals’ perceptions and recommendations regarding adolescent vaccinations in Georgia and Tennessee during the COVID-19 pandemic: a qualitative research 

Dear Dr. Olusanya:

I'm pleased to inform you that your manuscript has been deemed suitable for publication in PLOS ONE. Congratulations! Your manuscript is now with our production department. 

Kind regards, 

on behalf of

Dr. Florian Fischer 

Academic Editor

PLOS ONE